# Identification of lead vacancy defects in lead halide perovskites

David J. Keeble [1✉], Julia Wiktor[2], Sandeep K. Pathak[3,7], Laurie J. Phillips [4], Marcel Dickmann[5,6], Ken Durose [4], Henry J. Snaith [3] & Werner Egger[6]

Perovskite photovoltaics advance rapidly, but questions remain regarding point defects: while experiments have detected the presence of electrically active defects no experimentally confirmed microscopic identifications have been reported. Here we identify lead mono-vacancy ($V_{Pb}$) defects in MAPbI$_3$ (MA = CH$_3$NH$_3^+$) using positron annihilation lifetime spectroscopy with the aid of density functional theory. Experiments on thin film and single crystal samples all exhibited dominant positron trapping to lead vacancy defects, and a minimum defect density of ~3 × 10$^{15}$ cm$^{-3}$ was determined. There was also evidence of trapping at the vacancy complex $(V_{Pb}V_I)^-$ in a minority of samples, but no trapping to MA-ion vacancies was observed. Our experimental results support the predictions of other first-principles studies that deep level, hole trapping, $V_{Pb}^{2-}$, point defects are one of the most stable defects in MAPbI$_3$. This direct detection and identification of a deep level native defect in a halide perovskite, at technologically relevant concentrations, will enable further investigation of defect driven mechanisms.

[1] Physics, SUPA, School of Science and Engineering, University of Dundee, Dundee DD1 4HN, UK. [2] Department of Physics, Chalmers University of Technology, SE-412 96 Gothenburg, Sweden. [3] Clarendon Laboratory, Department of Physics, University of Oxford, Oxford OX1 3PU, UK. [4] Stephenson Institute for Renewable Energy, Department of Physics, University of Liverpool, Liverpool L69 7ZF, UK. [5] Physics Department and Heinz Maier-Leibnitz Zentrum (MLZ), Technische Universität München, D-85748 Garching, Germany. [6] Institut für Angewandte Physik und Messtechnik, Universität der Bundeswehr München, D-85579 Neubiberg, Germany. [7] Present address: Centre for Energy Studies, Indian Institute of Technology Delhi, New Delhi 110016, India. ✉email: d.j.keeble@dundee.ac.uk

The unprecedentedly rapid development of metal halide perovskite materials has been driven by their favorable optoelectronic properties enabling a new photovoltaic (PV) solar cell technology[1], and creating potential for other optoelectronic device applications[2,3]. Metal halide perovskites have a crystal structure (Fig. 1) with the general formula $ABX_3$, where the B-site metal cation is octahedrally coordinated by X-site halogen anions. In the subgroup of halide perovskites termed hybrid perovskites the more open corner A-sites are occupied by an organic molecule cation. Methylammonium (MA) lead iodide, $MAPbI_3$, is the prototypical metal halide hybrid perovskite. Understanding the origin of the high PV conversion and light emission efficiency remains a central aim of halide-perovskite research[2,4]. Recently there has been recognition of the importance of defect-assisted carrier recombination via centers which can act to limit the conversion efficiency of solar photovoltaic devices. Ultimately it is necessary to identify and confirm the point defect types present in order to design manufacturing processes for optimum performance. However, since experimental identification of point defects requires the use of spectroscopic methods that provide direct local structural information, or laborious studies to correlate for example electrical measurements across a sequence of chemically controlled sample sets, the vast majority of our present understanding comes from computational research effort[4–9]. Experimental studies, the subject of this work, are therefore urgently required. Nevertheless, first-principles calculations are providing detailed insight on the possible point defects and their role in fundamental mechanisms responsible for the intriguing materials physics of halide perovskites[5,9–13].

Currently our detailed understanding of the point defect physics and chemistry of hybrid perovskites therefore results exclusively from first-principles calculations. Using hybrid functionals and accounting for Spin-Orbit Coupling (SOC)[10,11,14] these methods then yield accurate values for the energies of the band edges and the band gap energy which is in the range 1.58–1.60 eV, in good agreement with the experimental value. When combined with appropriate schemes to correct for localized charge within the supercell these approaches enable defect formation energies (DFE) and charge transition levels to be calculated[10,13]. Studies of the primary native defects in $MAPbI_3$, the three monovacancy, $V_{MA}$, $V_{Pb}$, $V_I$, the interstitial defects, $MA_i$, $Pb_i$, $I_i$, and the two relevant antisite defects, $I_{MA}$, $Pb_I$, have been performed[5,10,11,13]. Point defects with charge transition levels deeper into the band gap are of particular relevance for defect-assisted mechanisms[2,4,6–9], such as nonradiative recombination[4,11,13]. While $I_{MA}$ is reported to have a (0/2−) charge transition level deep in the band gap[13], the antisite defects were found to have high DFE values so are expected to have negligible concentrations[11]. The cation interstitial, $Pb_i$ and $MA_i$, yield shallow donor charge transition levels close to the conduction band minimum (CBM)[11]. By contrast, the iodine interstitial was found to be a deep defect and to have one of the lowest DFE values, and so be one of most stable defects[11,13]. It exhibits a (+/−) transition level 0.95 eV above the valence band maximum (VBM), the neutral charge state is energetically less favorable than either $I_i^+$ or $I_i^-$. The trapping of two carriers occurs rapidly, but sequentially via the two associated charge transition levels, (0/−) and (+/0) at 0.78 eV above the VBM and 0.58 eV below the CBM, respectively[13]. The center can enable nonradiative carrier recombination and to quantify the recombination rate both the electron and hole capture rates are required. Recent first-principles calculations conclude that the iodine interstitial is the primary nonradiative recombination center in hybrid perovskites[13].

The anion vacancy, $V_I$, is reported to have a shallow donor charge transition level close to the CBM and hence to be normally stable in the positive charge state[5,10,11], $V_I^{1+}$. The A-site cation (MA) vacancy forms shallow acceptor charge transition level close to the VBM so is normally negatively charged, $V_{MA}^{1-}$, but the relatively high DFE imply concentrations are low[10]. It has recently been shown that the MA vacancy is expected to be noticeably more stable on MAI terminated surfaces[5]. The B-site cation vacancy, the lead vacancy, $V_{Pb}$, is a deep acceptor defect with a (0/2−) charge transition level 0.5 eV above the VBM[5,10,11,13]. Again, trapping of carriers is expected to proceed sequentially via the shallow (−/2−) level, 0.13 eV above the VBM, and a deep (0/−) level, ~0.8 eV above the VBM. Nonradiative recombination is controlled by the slowest carrier capture process, which should be electron capture by the (−/2−) level, and some first-principles conclude that $V_{Pb}$ might make a limited contribution. The double negative charge state, $V_{Pb}^{2-}$, is calculated to have small DFE values under all growth conditions and so is

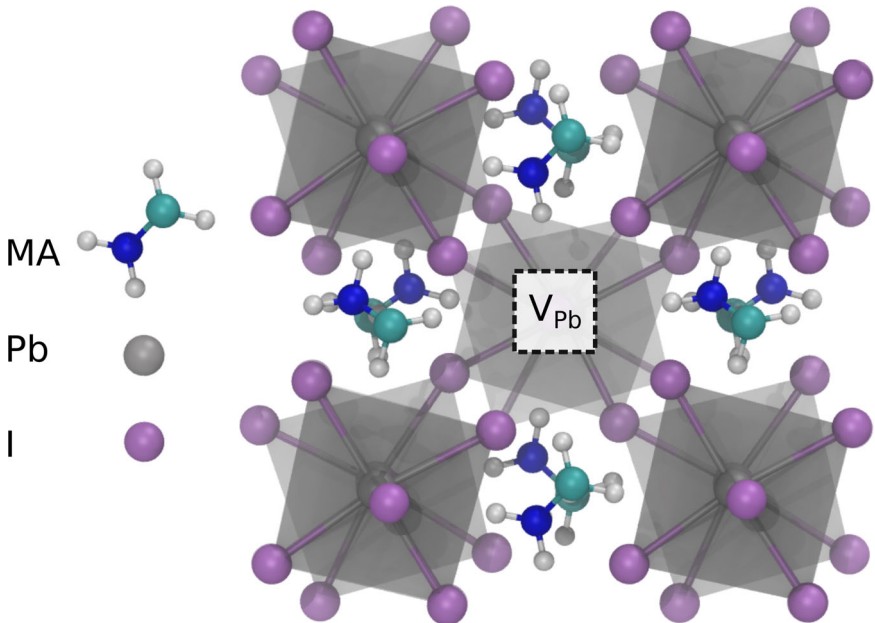

**Fig. 1 MAPbI₃ crystal structure.** Illustrating B-site polyhedra (gray), and showing a Pb vacancy.

one of the most stable defects in MAPbI$_3$[9–11]. It has been proposed that for low Fermi energies $V_{Pb}^0$ will decompose following the reaction[10], $V_{Pb}^{2-} + 2h^+ = V_{Pb}^0 = V_{Pb}^{2-} + I_i^+ + V_I^+$. Recent calculations for PbI$_2$ terminated surfaces predict $I_i$, $V_{Pb}$ and $V_I$ are more stable, and that the $V_{Pb}$ (0/2–) level, and associated (0/–) and (–/2–) transitions, move deeper into the band gap due to the increased stability of $V_{Pb}^0$ implying an increased contribution to nonradiative recombination[5].

Deep-level transient spectroscopy (DLTS)[15–17], and thermally stimulated current[18], measurements clearly show the presence of deep-level defects in MAPbI$_3$. Defect concentrations have been inferred from DLTS[15], and space-charge limited current measurements (SCLC)[19], and have been reported to be on the order of $10^{17}$ cm$^{-3}$ or lower[7], but caution in the interpretation of SCLC measurements is required[20]. Furthermore, while DLTS is capable of identifying charge transition energy levels of defects in the gap, it does not of itself give any indication of the chemical identity of point defects in the lattice. Although in principle it should be possible to correlate the experimental energies with those from density functional theory (DFT) calculations, in practice this is challenging. Deep defect energy levels have been reported from a DLTS study of thin film MAPbI$_3$ at CBM – 0.62(1) eV and at CBM – 0.75(1) eV[15], and at VBM + 0.84(1) eV from an optical DLTS study of crystal MAPbI$_3$[16]. Both studies referred the experimental defect charge transition energies to the values calculated by Yin et al.[21] using the projector augmented wave (PAW) with general gradient approximation (GGA) DFT method and including SOC. The thin film results were attributed to $I_{Pb}$ antisite and the $I_{MA}$ antisite on this basis[15], while the trap level observed from the crystal study was associated with $I_{Pb}$[16]. However, the point defect charge transition energy level positions within the gap calculated using PAW-GGA-SOC[21] differ markedly from those obtained in more recent calculations using hybrid functionals and including SOC[10,11,13]. For example, the lead vacancy was predicted to give a single shallow acceptor (–/2–) level in the gap as opposed a deep (0/2–) acceptor level as determined by the hybrid functionals with SOC calculations. While the resulting defect geometries, in particular for defects in their formal charge states, e.g., $V_{Pb}^{2-}$ or $V_{MA}^-$, are typically similar for the two calculation schemes, the use of hybrid functionals with the inclusion of SOC and appropriate schemes to correct for localized charge are of critical importance for the determination of charge transition energy level position. The stark differences in the energy level values highlight current uncertainties regarding point defect identification. Furthermore, it has been shown that the set of trap energies detected by DLTS and attributed to the perovskite layer can depend on the partner hole transporting material in the device configuration[22]. The presence of grain boundary potentials and traps can also complicate the interpretation of DLTS experiments[23].

Positron annihilation spectroscopy methods yield information that depends on the local structure of the defect localizing the positron and can provide identification of neutral or negatively charged vacancy–related defects in materials[24–26]. Positron annihilation lifetime spectroscopy (PALS) enables the experimental detection of multiple positron states. Observation of both perfect lattice annihilation states and longer lifetime defect positron states enables the trapping to vacancy-related defects to be unambiguously established. Comparison of the experimentally determined lifetimes with the DFT calculated values for the perfect material state and of states localized at specific vacancy–related defects[26,27] allows the identification of specific point defect types. Moreover, positron lifetime measurements performed using a high intensity positron beam enable depth–profiling of vacancy–related defects from the near-surface down to depths of a few microns[28,29].

Here we report the detection and identification of lead vacancy related defects in thin film, and in the near surface region of single crystal MAPbI$_3$ using variable positron implantation energy PALS measurements performed with the high intensity positron beamline (NEPOMUC) at the Heinz Maier–Leibnitz Zentrum (MLZ) research reactor in Garching[28,29]. Calculations of positron lifetimes using both atomic superposition and projector augmented-wave (PAW) DFT methods for perfect lattice MAPbI$_3$ and relevant vacancy defects are performed. Positrons localize at missing atom defects with open volume, however, for this to occur the local charge of the defect is required to be neutral or negative with respect to the lattice; the trapping rate to positively charged vacancy-related defects is negligible[25,26]. The lifetime of positrons localized at vacancy defects is a measure of the local electron density, and is a characteristic of the specific vacancy defect type. Similarly, the positron lifetime resulting from annihilation of positrons delocalized in Bloch states in the perfect material, the bulk lifetime, is characteristic for a given material. The vacancy defect positron lifetimes are greater than the bulk lifetime. The primary vacancy defects in MAPbI$_3$ are the anion iodine vacancy, $V_I$, and the two cation vacancies (the larger A-site MA-ion, CH$_3$NH$_3^+$, vacancy, $V_{MA}$, and the B-site, octahedrally coordinated, Pb vacancy, $V_{Pb}$) as shown in Fig. 1. Positrons are expected to trap at either cation vacancy, i.e., the Pb vacancy with local charge of –2 or the MA ion vacancy with a local charge of –1, but will not trap at positively charged anion vacancies.

## Results

**Density functional theory calculations of the characteristic positron lifetime values.** The positron annihilation lifetime values were calculated for annihilation from perfect lattices states, and for states localized at the two negatively charged cation vacancies. They were performed using the atomic superposition method, implemented in the program MIKA-Doppler[30,31], and with the PAW method using ABINIT[27,32]. ABINIT calculations were also performed were the neighbor atoms to the vacancy relaxed according to the forces due to the electron and positron densities (further details of the calculations are given in the theoretical calculations sections of Methods and Supplementary Note 1). The resulting calculated positron state lifetime values for MAPbI$_3$ are given in Table 1. There is approximate agreement between the two calculation methods. The low electron density in MAPbI$_3$ resulted in a long perfect lattice (bulk) positron lifetime in the approximate range 340–350 ps. The positron lifetime for the perovskite B-site, Pb, vacancy was found to be ~369 ps. The resulting positron density localized at a Pb vacancy is shown in Fig. 2. The lifetime value for the more open A-site, MA ion, vacancy was in the approximate range 400–440 ps. Calculations were also performed for the possible A-site vacancy, $V_{MA}$, iodine interstitial complex $(V_{MA} - I_i)^{2-}$ and for the possible B-site centered divacancy, $(V_{Pb}V_I)^-$. Both yielded lifetime values greater

**Table 1 Calculated positron state lifetimes (ps) for perfect lattice and cation vacancy defects in MAPbI$_3$.**

| Positron state | MIKA | ABINIT | ABINIT-relaxed |
|---|---|---|---|
| Perfect lattice | 353 | 342 | |
| $V_{Pb}^-$ | 369 | 360 | 369 |
| $(V_{Pb}V_I)^-$ | | | 377 |
| $V_{MA}^-$ | 401 | 414 | 442 |
| $(V_{MA} - I_i)^{2-}$ | | | 403 |

Atomic superposition calculations were performed with MIKA-Doppler, PAW-DFT calculations with ABINIT. ABINIT calculations were also performed relaxing the structure in the presence of the positron.

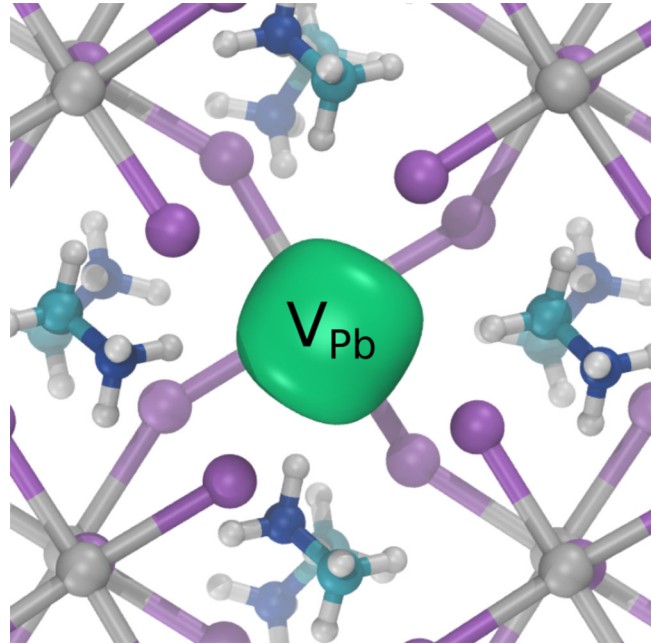

**Fig. 2 Calculated positron density at the Pb vacancy in MAPbI₃.** Positron density isosurface shown in green localized at a $V_{Pb}^{2-}$ defect obtained using ABINIT relaxing the structure in the presence of the position.

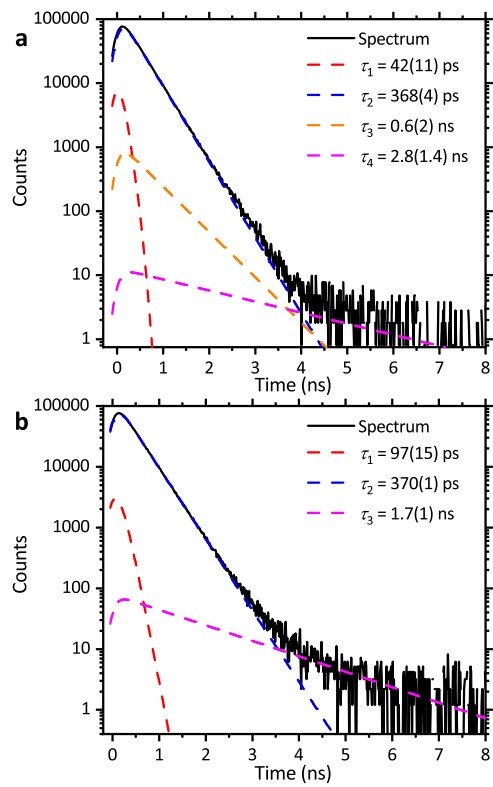

**Fig. 4 Experimental positron lifetime spectra with deconvolution fit components. a** MAPbI₃ crystal with 4 keV implantation energy, both also showing the fitted positron lifetime components. **b** The 300 nm MAPbI₃ film (Liverpool, Anneal A) measured with 2.5 keV positron implantation energy.

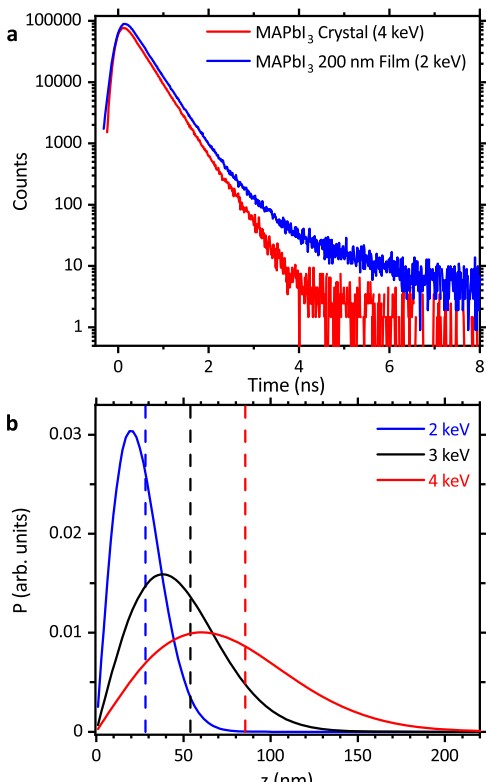

**Fig. 3 Experimental positron lifetime spectra and positron implantation profiles. a** Lifetime spectra for a 200 nm MAPbI₃ film (Oxford), (blue) measured with 2 keV positron implantation energy and for a MAPbI₃ crystal (red) at 4 keV. **b** Markovian positron implantation depth profiles, the mean depths are shown with dashed lines.

than, and separated from, the Pb vacancy lifetime but were shorter than the $V_{MA}$ value (Table 1). A similar trend in positron lifetime values for the two cation vacancies has been observed in ABO₃ perovskite oxide materials with a bulk lifetime in the 150–160 ps range, the $V_B$ lifetime typically 180–200 ps, and the $V_A$ lifetime in the 280–300 ps range. Both cation vacancy defects have been experimentally observed[24,33].

**Variable energy positron annihilation lifetime spectroscopy measurements**. These were performed on MAPbI₃ thin films from both Oxford (200 nm) and Liverpool (300 nm), and a single crystal grown by inverse temperature crystallization. Figure 3a shows positron lifetime spectra from a thin film and from the near surface of the single crystal while Fig. 3b shows positron implantation profiles for the implantation energies. Positron lifetime spectra were deconvolved to identify the contributing positron state components, see Fig. 4 and Table 2 (complete results are given in Supplementary Note 2). A dominant positron lifetime component with intensity greater than or equal to 92% and an average lifetime value of 370(3) ps was obtained from the spectra measured from all five of the MAPbI₃ samples studied.

The first component lifetime value (Table 2, Supplementary Note 2) was significantly smaller than the DFT calculated perfect lattice (bulk) lifetime value (Table 1). This reduced bulk lifetime component has low intensity, nevertheless, the experimental lifetime component values can be used to calculate a standard trapping model bulk lifetime value of 342(16) ps (see Supplementary Note 3) which is in agreement with the DFT results (Table 1). The observation of a dominant positron lifetime component (370(3) ps) with a value greater than the experimental

**Table 2 Experimental positron lifetime component results for MAPbI$_3$ thin film and single crystal samples.**

| Sample | Source | E (keV) | $\tau_1$ (ps) | I$_1$ (%) | $\tau_2$ (ps) | I$_2$ (%) | $\tau_3$ (ns) | I$_3$ (%) | $\kappa_2$ (s$^{-1}$) |
|---|---|---|---|---|---|---|---|---|---|
| Film | Oxford | 2 | 119(45) | 2.2(8) | 371(5) | 94(1) | 0.7(1) | 3.5(1.6) | 5.6(2.2) x 10$^9$ |
| Crystal | Dundee | 4 | 42(10) | 4.6(5) | 368(4) | 94(3) | 0.6(3) | 1.5(2.5) | 2.0(5) x 10$^{10}$ |
| Film (Standard) | Liverpool | 2.5 | 109(19) | 2.7(3) | 375(1) | 97.3(2) | 2.3(4) | 0.05(1) | 6.3(1.1) x 10$^9$ |
| Film (Anneal A) | Liverpool | 2.5 | 49(27) | 2(1) | 367(1) | 98(1) | 1.4(1) | 0.29(4) | 7.5(1.2) x 10$^9$ |
| Film (Anneal B) | Liverpool | 2.5 | 110(24) | 2.3(3) | 370(1) | 97.4(3) | 1.7(1) | 0.27(3) | 1.7(9) x 10$^{10}$ |

Deconvolved component lifetime values, $\tau$, intensities, I, calculated defect position trapping rates, $\kappa_2$, and the positron implantation energies, E.

standard trapping model calculated bulk lifetime provides direct evidence for trapping to vacancy-related point defects with neutral or negative local charge in all the MAPbI$_3$ samples studied.

Comparing the dominant component experimental lifetime of 370(3) ps with the DFT calculated lifetimes for the two cation vacancies, $V_{MA}^{1-}$ and $V_{Pb}^{2-}$, there is very good agreement with Pb vacancy value of ~369 ps (Table 1). By contrast, the calculated positron lifetime value characteristic of the more open twelve coordinated CH$_3$NH$_3^+$ site (A-site) is greater than 400 ps. A more detailed consideration of the dominant lifetime component values shows that for four of the five MAPbI$_3$ samples studied the lifetime was in the range 369(2) ps while for the other sample this was 375(1) ps (Supplementary Note 2). The DFT calculated lifetime for the $(V_{Pb}V_I)^-$ divacancy defect is 377 ps (Table 1) suggesting the possibility of trapping to both $V_{Pb}^{2-}$ and $(V_{Pb}V_I)^-$ defects. A very low intensity experimental component with a lifetime in the approximate range 600–800 ps was also detected from one of the thin film samples, and from the near surface region of the single crystal (Table 2, Supplementary Tables 1 and 2), is due to the annihilation of positronium (see Supplementary Note 4) and cannot be attributed to a specific defect type.

The observation of lead vacancy defects, and the absence of trapping to MA vacancies, in the studied MAPbI$_3$ samples is consistent with first-principles study predications that $V_{Pb}^{2-}$, in contrast to $V_{MA}^{1-}$, exhibits low DFE values under all growth conditions and so is one of the most stable defects. First principles calculations have provided evidence that lead vacancies are deep level defects capable of trapping holes[9–11].

### Estimation of lead vacancy concentration.
The detection of a weak reduced bulk lifetime component in all the measured spectra enables the rate of positron trapping to vacancy defects, $\kappa_D$, to be estimated (see Table 2 and Supplementary Note 2). The vacancy defect concentration is related to the trapping rate, $[V] = \kappa_D/\mu_V$, where $\mu_V$ is the defect specific trapping coefficient. The values for $\mu_V$, for negatively charged vacancy defects in various semiconductors, have been reported to be in the range[25] ~$1 \times 10^{15}$–$3 \times 10^{16}$ s$^{-1}$. In consequence, lead vacancy defect densities greater than ~$3 \times 10^{15}$ cm$^3$ are required to yield the experimental defect lifetime component intensities observed from all the MAPbI$_3$ samples measured. Using the more typical value of $\mu_V$ for negatively charged vacancies of ~$2 \times 10^{15}$ s$^{-1}$ an average value of defect densities obtained for the samples studies is estimated to be $9(6) \times 10^{16}$ cm$^3$ (see Supplementary Note 3).

### Discussion
Our work demonstrates the experimental detection and identification of native vacancy point defects in thin film and single crystal metal halide perovskite materials. Positron annihilation lifetime spectroscopy, with the aid of two-component DFT calculations, enables the identification of cation vacancy and vacancy cluster defects in MAPbI$_3$. In all the thin film and crystal samples studied a dominant, ≥92% intensity, positron trap with a positron state lifetime of 370(3) ps (Table 2, and Supplementary Note 2) consistent with the DFT calculated value of 369 ps (Table 1) for $V_{Pb}$ was observed. The possible trapping to $(V_{Pb}V_I)^-$ divacancy defects was observed for one of the thin film samples. The lead vacancy related defect density was found to be greater than ~$3 \times 10^{15}$ cm$^{-3}$ in all samples. Using a more typical value for the positron trapping coefficient for a negatively charged vacancy defect yields an average defect density of $9(6) \times 10^{16}$ cm$^3$ for the samples studied. No positron trapping to MA vacancy defects was detected. Our results support the predications of first-principles calculations that deep level, hole trapping, $V_{Pb}^{2-}$ point defects are one the most stable defects in MAPbI$_3$ and that MA vacancies are expected to have negligible concentrations. The results are also in agreement with recent low-dose scanning transmission microscopy studies of a metal halide perovskite that provide evidence for the presence of vacancy defects in Pb-I sublattice[34]. Depth-profiling positron lifetime spectroscopy is demonstrated to be a point defect characterization method that can be applied to metal halide perovskites, enabling the detection and identification of neutral or negatively charged vacancy-related defects.

### Methods
**Sample preparation.** The University of Oxford MAPbI$_3$ films were formed from 40 wt% precursor solution of MAI:PbI$_2$ dissolved in dimethlformamide (DMF). The MAI was made in house[35]. The precursor solution was coated on a clean FTO coated glass sheets by a consecutive two-step spin-coating process at 1200 rpm and 2000 rpm for 25 s and 12 s, respectively. The films were annealed at 150 °C for 15 min. Two films were top coated with 50 nm PMMA in chlorobenzene. All processing was performed in a nitrogen-filled dry glovebox. All materials were purchased from Sigma-Aldrich or Alfa Aesar and used as received.

The University of Liverpool MAPbI$_3$ films were synthesised using a one-step solution by a DSMO adduct with antisolvent method. A 1:1:1 MAI:PbI$_2$:DMSO in DMF solution was spin-coated, with chlorobenzene added during the spin process removing the DMF and allowing rapid conversion to MAPbI$_3$ on thermal annealing. This produced a standard film, for the anneal A and B films a small additional volume of DMSO, 40 mL and 80 mL, respectively, was present during the anneal. The films were made and packaged entirely in a glovebox, then vacuum sealed. All chemicals were supplied by Sigma-Aldrich with the exception of MAI, which was purchased from Solaronix SAs. All materials were used as received, without further purification processes.

The MAPbI$_3$ crystals measured were grown by the inverse temperature crystallization procedure[36]. 1 M solutions of PbI$_2$ (99%, Sigma-Aldrich) and MAI (Dyesol Ltd) were prepared in $\gamma$-butyrolactone (>99% Sigma-Aldrich), the solutions were filtered using PTFE filter with 0.2 μm pore size. The filtered solution mixture was placed in a flat-bottomed glass vial and placed in a silicone oil bath maintained at 110 °C. All procedures where carrier out in ambient conditions and 20–30% relative humidity. Crystal structure was confirmed by x-ray diffraction using a Siemens D5000 diffractometer. The crystals exhibited rhombo-hexagonal dodecahedra morphology and had a smooth major facet size of ~$8 \times 9$ mm$^2$.

**Positron annihilation.** Variable energy positron annihilation lifetime spectroscopy was performed using the PLEPS instrument on the NEPOMUC high intensity beam line at the Heinz Maier–Leibnitz Zentrum (MLZ) research reactor in Garching[28,29]. Spectra contained $4 \times 10^6$ counts. The spectra were fitted using the software package PALSfit Version 3.195 (Technical University of Denmark, Riso Campus)[37]. The timing instrument resolution function was determined using a SiC standard sample, was described by three Gaussian functions, the resulting full width half maximum values varied from 277 ps to 312 ps for the measurements

described. Samples were transferred from an inert gas environment container to the sample chamber and a vacuum established within several minutes. Measurements were also performed on $CH_3NH_3PbI_3$ thin films with a thin PMMA environment protecting coating and the results were found to be consistent with those obtained from non-coated films (see Supplementary Note 5).

**Theory calculations**. Positron lifetime calculations were performed using atomic superposition method with the MIKA-Doppler code[31]. In the calculations we applied the electron-positron enhancement factor[30,38] resulting from the parametrization of the data of Arponen and Pajanne[39]. Supercells of $CH_3NH_3PbI_3$ comprising 1152 atoms were used for the computationally efficient atomic superposition calculations[40]. The projector augmented-wave (PAW) method calculations were carried out using the ABINIT code[27,32] with the electron-positron correlation functional parametrized by Boronski and Nieminen[41]. The electronic and positronic densities were updated self-consistently in a double loop. During each subloop, one of the two densities was kept constant while the other was being converged. To account for the imperfect screening of the positron by the electrons in MAPbI3, we applied the gradient correction proposed by Barbiellini et al.[30]. The generalized gradient approximation (GGA) as parametrized by Perdew, Burke, and Ernzerhof (PBE) was used to describe the electron exchange-correlation interactions[42]. The positron lifetime calculations were based on a 192-atom supercell with an initial apolar arrangement of the organic cations. The energy cutoff was set to 16 Ha. We used lattice parameters corresponding to the tetragonal phase of $CH_3NH_3PbI_3$ ($a = b = 1.772$ nm, $c = 1.266$ nm)[43], and a $1 \times 1 \times 2$ Monkhorst-Pack $k$-point mesh. The defects were relaxed using forces due to both electron and positron densities until convergence below 1 ps was reached. Additional tests were performed to assess the effect of spin orbit coupling and the use of hybrid functionals on the calculated positron lifetimes (see theoretical calculations section of the Supplementary Note 1). It was found that including SOC reduced the calculated positron lifetime but that using hybrid functionals increased this value and hence the combination resulted in a lifetime in agreement with the GGA without SOC value. Hence the supplementary calculations confirmed the validity of the approach used.

## Data availability

The positron lifetime data that support the findings of this study and source data for display items has been deposited in figshare with the identifier doi:10.6084/m9.figshare.15031926.

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

## Acknowledgements

D.J.K. gratefully acknowledges the financial support provided by FRM-II to perform the high-intensity positron beam measurements at Heinz Maier-Leibnitz Zentrum (MLZ), Garching, Germany. J.W. acknowledges funding from the "Area of Advance—Materials Science" at Chalmers University of Technology and the Swedish Research Council (2019–03993). The computations were partly performed on resources provided by the Swedish National Infrastructure for Computing (SNIC) at NSC and PDC. L.J.P. and K.D. would like to acknowledge support from EPSRC grants EP/N014057/1 m and EP/M024768/1. H.J.S. acknowledges support from EPSRC grant EP/L024667/1. W.E. gratefully acknowledges BMBF-grants 05K13WN1-POSIANALYSE, 05K16WN1-POSITEC and 05K19WN1-POSILIFE of the German Federal Office of Research and Education.

## Author contributions

D.J.K. with S.K.P. and L.J.P. designed the study. S.K.P., H.J.S., L.J.P., K.D., and D.J.K. supplied samples. D.J.K., M.D. and W.E. performed the positron annihilation experiments. J.W. performed PAW-DFT calculations and D.J.K. the atomic superposition calculations. D.J.K., with help from J.W. and K.D., wrote the main draft. All authors commented on the manuscript.

## Competing interests

The authors declare no competing interests.
