## [Peer Review File · Nature Communications]

Identification of lead vacancy defects in lead halide perovskitesREVIEWER COMMENTS

Reviewer #1 (Remarks to the Author):

This manuscript reports provides key experimental insights into the nature of point defects in organic-inorganic lead halide perovskites. Positron lifetime spectroscopy and first principles calculations were applied skillfully to identify for the first time the presence of vacancies that involve the lead cation vacancy in MAPbI₃ thin films and in a single crystal of MaPbI₃.

Such important findings deserve publication in Nature Communications, as the relevance for both the fundamental understanding and the development of perovskite solar cells is huge. The current manuscript will enable and inspire systematic follow-up studies on the quantitative relationship between point defects and non-radiative recombination of charge carriers – a key process that as of today hampers to achieve optimum efficiencies in the development of solar cells based on MAPbI₃ and related hybrid organic-inorganic perovskites.

Nevertheless, I have a major comment and some minor comments to the manuscript in its current state that need consideration before publication can be recommended.

Main comment

1. At several points in the body of the manuscript, the authors claim that the dominant vacancy observed is the V_{Pb2-} cation vacancy. However, the first-principles calculations show that the positron lifetime for the (V_{Pb}-V_I)- divacancy (377 ps) is very close to that of the lead vacancy (369 ps). Both calculated positron lifetimes are in agreement with the observed range of positron lifetimes, namely 367 to 375 ps, as the uncertainty in the absolute value of both experimental and theoretical positron lifetimes is typically around 5 ps. Therefore, this study demonstrates that either V_{Pb2-} vacancies or (V_{Pb}-V_I)- divacancies, or a combination of both species, are present in the systems investigated. This logical deduction and major conclusion should be clearly stated throughout the manuscript instead of only referring to the lead vacancy VPb₂₋. Please note that the formation of V_I vacancies in MAPbI₃ is relatively easy, therefore the formation of such cation-anion divacancies may well occur in abundant concentrations.

Minor Comments

- A single value for the rate of positron trapping to vacancy defects, KD, estimated to be $4.2 \times 10^9 \text{ s}^{-1}$ is stated. But Eq. (S2) shows that KD will vary systematically with the observed reduced bulk lifetime that ranges between 40 ps and 150 ps, depending on sample and positron implantation energy (probe depth). It is advised to specify KD for all the analyzed lifetime spectra in the respective tables, and use the spread to estimate a variation interval for KD, and not only to specify the estimated average. The spread is expected to reflect (and may thus give insights into) the variation in concentration of the vacancy defects between samples and with depth within the film.

- As the authors state "The VPb₂₋ cation vacancy is a deep acceptor defect that leads to trapping of carriers via the shallow (-/2-) level, 0.13 eV above the VBM, and a deep (0/-) level, ~0.8 eV above the VBM." They also refer to DLTS, as it is a very sensitive technique to detect the presence of point defects and provides insights into the energy density of states within the band gap of lead halide perovskites. A comparison of the main results and conclusions of the current manuscript with those from DLTS studies on MAPbI₃ as reported in the literature is needed. Is the lead vacancy also observed by DLTS, are the energy levels quoted for the lead vacancy consistent with observations by DLTS? What about the defect levels for the (V_{Pb}-V_I)- divacancy, what are the involved energies within the band gap, and are such divacancies observed by DLTS, or not?

Reviewer #2 (Remarks to the Author):

The manuscript reports the detection and identification of the lead monovacancy (VPb) defects in MAPbI₃ (MA = CH₃NH₃⁺) using Positron Annihilation Lifetime Spectroscopy with the aid of Density Functional Theory (DFT). The results support the predictions of other first-principles studies that deep level, hole trapping, VPb–2 point defects are one of the most stable defects in MAPbI₃.

The Positron Annihilation Lifetime Spectroscopy is a well known technique for the detection of point defects in semiconductors, and the DFT method is also a well known theoretical method for predicting defect properties, with dozens of papers about DFT studies on point defects in MAPbI₃ published in the past decade. The present work combines the two techniques and supports the predictions of previous studies (e.g., Energy Environ Sci11, 702-713 (2018)). The novelty and significance are not so obvious to me. The results should be published in a more specialized journal.

The authors mentioned that the defect calculations usually require use of hybrid functionals and must account for spin-orbit coupling. In fact, the use of hybrid functionals and the spin-orbit coupling are very important for predicting the properties of Pb vacancy VPb, as reported in Energy Environ Sci11, 702-713 (2018).

Unfortunately, this is not taken into account in the present study. As mentioned in the method section, the generalized gradient approximation (GGA) as parametrized by Perdew, Burke, and Ernzerhof (PBE) was used to describe the electron exchange-correlation interactions. I do not understand why the important hybrid functionals and the spin-orbit coupling are ignored.

There are typos in many sentences, even in the abstract, e.g., Experiments on thin film and single crystal samples and all exhibited a dominant positron trapping to lead monovacancy defects and a minimum defect density of $\sim 3 \times 10^{15} \text{ cm}^{-3}$ was determined.

Reviewer #3 (Remarks to the Author):

Keeble et al. report on a combined experimental and theoretical effort aimed at investigating the nature of point defects in MAPbI₃. While this topic has received a large attention in the literature, the use of positron annihilation lifetime spectroscopy shed fresh light on the subject and allows to clearly identify the lead vacancy as the dominant vacancy defect in the material.

The manuscript is generally well-written and the results are carefully explained.

While I surely support the publication of the manuscript as is, I would like the authors to comment on a few questions, reported below.

1) the authors have focussed on bulk properties, while it appears that the features near the surface cannot be unambiguously attributed. Would further experiments be able to provide vital information also on the nature of surface defects or the shortcomings here observed cannot be overcome?

2) Could the authors explain more clearly the computational details? It is not clear which calculation use the 1152-atoms supercell and which the 192-atoms supercell and the motivation beyond the computational protocol.

3) Furthermore, could the author explain if and how the use of a GGA functional without SOC, instead of the hybrid functional+SOC usually employed for calculations of defects in MAPbI₃, influences the calculated results? Is it possible that the nice agreement observed between experimental and calculated lifetimes originates from error cancellation in the computed values?

4)The authors state that a minimum defect density of $\sim 3 \times 10^{15} \text{ cm}^{-3}$ was determined but I would find nice for consistency to have the data for each sample listed as done for other quantities. Could the authors provide all the data in a Table for the different systems in SI?

Reviewer #1

Reviewer #1 articulates the importance of work detailed in our manuscript and affirms that it is worthy of publication in Nature Communications.

The **main comment** made by reviewer #1 on the manuscript related to the closeness of the calculated positron lifetime values for the monovacancy defect at 369 ps with that for the divacancy defect comprising a lead vacancy with a nearest neighbour iodine vacancy at 377 ps, when compared to the dominant experimental positron lifetime value which is in the range 367 ps to 375 ps.

We have amended our description of the results (top of page 14) to point out that four of the five samples study return a defect lifetime in the range 367 ps – 371 ps, while one of the thin films yields 375(1) ps (see Supplemental Tables 1 to 5). We agree that while trapping to V_{pb}^{2-} vacancies is predominant, trapping to $(V_{\text{pb}}V_{\text{I}})^{1-}$ divacancies, or both may occur and we amended the manuscript to use the term '*lead vacancy related defects*' where appropriate. However, we consider the difference between the defect lifetime observed from eight of the ten fits between 367 ps – 371 ps giving an average of 369.8(1.4) ps and the calculated divacancy lifetime of 377 ps lie outside the boundary of probable error and hence we also retain the headline identification of the lead monovacancy defect.

A **minor comment** was made that we should report the rate of positron trapping to the defect, κ_D , for each of the spectra analysed.

We agree with the reviewer and we have added these values to Table 2 and to Supplementary Tables 1 – 5 and have expanded the related section on 'Standard Trapping Model defect concentration calculations' within the Supplementary Information. We have also given values for the spread in estimated defect concentrations across the samples studied using the most typical value for the trapping coefficient for negatively charged vacancies, see bottom of page 14. In practice for these samples the variations in lead vacancy related defect concentrations do not provide significant insight. For some samples this concentration is slightly higher nearer the surface, for others the reverse is

true. We have now provided all the necessary information so an interested reader can easily compute and evaluate these behaviours.

A second **minor comment asks** whether our results are supported by DLTS and focused on a statement in our manuscript, “The B-site cation vacancy, the lead vacancy, V_{Pb} , is a deep acceptor defect with a $(0/2-)$ charge transition level 0.5 eV above the VBM^{5,10,11,13}. Again, trapping of carriers is expected to proceed sequentially via the shallow $(-/-2-)$ level, 0.13 eV above the VBM, and a deep $(0/-)$ level, ~ 0.8 eV above the VBM.” It is possible the reviewer was misled into thinking this was our work, however, we are simply quoting the relevant literature (as referenced) in these two sentences. Similarly our discussion of the relevant DLTS literature was contained in the introductory paragraphs. The work we performed was restricted to experimental positron annihilation measurements and specialised two-component density function theory calculations of positron states and their lifetimes. We do not perform DLTS measurements and we do not perform the type of first principles calculations that enable charge transition levels to be calculated. In consequence, the questions asked by the reviewer cannot be addressed. We would also point out that DLTS is not capable of identifying the microscopic nature of the electrically active defect responsible for a given DLTS peak. Further, we note that reliable DLTS measurements on MAPI are particularly challenging to perform.

We have added a substantive paragraph in the revised manuscript to more clearly review the DLTS studies performed on MAPI on page 6 (also see above).

We fully agree with the reviewer that the questions raised are very interesting. We would indeed plan in future experiments to use both positron annihilation lifetime spectroscopy and DLTS, if these prove technically feasible, to investigate possible correlations that could enable an experimental identification of the charge transition level due to a specific vacancy defect. This is another example of the type of fundamental experiment that the current study will enable.

Reviewer #2

We agree with Reviewer #2 that we should have clearly explained our handling of spin-orbit coupling and of using hybrid functionals. We have added a section to the Supplementary Information for the revised manuscript that detail the computational results where both SOC and hybrid functionals are included. The net effect is to leave the calculated positron lifetimes essentially unchanged. Including SOC reduced the calculated lifetime for the lead vacancy by 6 ps, but then using hybrid functional increased the lifetime by 5 ps. The SOC tends to delocalise electrons in MAPI while hybrid functionals tend to localise them. Details of these tests have been added to a new theoretical calculations section in Supplementary Information and to the Method section (page 18) has been amended.

We cannot agree with reviewer’s comment regarding the novelty and significance of our results. Our work is an **experimental** atomic-scale identification of a point defect in a halide perovskite. The paper cited by the Referee, *Energy Environ Sci* 11, 702-713 (2018), details first principles calculations on the native point defects in $MAPbI_3$. It also presents the results of photoluminescence and transient absorption measurements that provide evidence of non-radiative decay due to defects. *However, optical measurements of this type do not provide a microscopic identification of the type of point defect involved.* In this work we report the unambiguous identification of a specific type of native point defect, the lead vacancy. It should be noted that there are a very large number of purely computational studies on point defects in halide perovskites in prestigious wide-audience journals. We are therefore convinced that the first direct experimental identification of point defect in these materials is also of interest to a general audience and deserves to be published in *Nature Communications*.

We thank the reviewer for pointing out the typographical errors – we have corrected them and clarified some of the sentences in the revised manuscript.

Reviewer #3

Reviewer #3 clearly details the importance of the work described and supports publication in Nature Communications.

We are asked to comment on four points and do so below:

1) “the authors have focussed on bulk properties, while it appears that the features near the surface cannot be unambiguously attributed. Would further experiments be able to provide vital information also on the nature of surface defects or the shortcomings here observed cannot be overcome?” We understand the reviewer is requesting an amplified informational discussion within this response, rather than changes to the manuscript.

The reviewer is indeed correct we deliberately measured spectra over the implantation energy range 2 – 4 keV (Fig. 3(a)) for the thin films to optimally implant into the ‘bulk’ of the thin film and avoid possible further complexity within the surface and interface regions in the initial primary study.

We suspect that the referee is alluding to the statement within the manuscript “A very low intensity experimental component with a lifetime in the approximate range 600 – 800 ps was also detected from one of the thin film samples, and from the near surface region of the single crystal (Table 2, Supplemental Tables 1 and 2), is due to the annihilation of positronium (see Supplementary Information) and cannot be attributed to a specific defect type. ”. Firstly, the weak positronium component is observed in the 4 keV (mean implantation depth 85 nm), lowest energy, spectrum from the crystal and also in the both the 2/2.5 keV and 4 keV spectra from the thin films. In other words it is not restricted to the surface, for the films it has approximately equal intensity at both mean implantation depths studied (~30 & 85 nm). It can be noted that all the spectra regardless of implantation depth in fact exhibit a weak > 1 ns lifetime component with an intensity in the range 0.05 – 0.75 %. This can be attributed to a tiny concentration of larger open-volume nanovoids almost invariable detected in most materials and general considered to not be of great interest. However, in the 4 keV spectrum from the crystal and the two spectra (2 & 4 keV) from the Oxford films (Supplementary Table 1 and 2) there is a 0.6 – 0.8 ns component with an intensity in the range 1-5 %. This is also a positronium component, but in contrast to the > 1 ns component this cannot currently be easily interpreted. This situation is described and referenced in the ‘Positronium formation and annihilation’ section of the Supplementary Information. Further discussion of this rather localised positronium state would involve detailed and specialised positron physics and would not advance the readers understanding of the materials physics of MAPI. Some of the authors are interested in this point and the result adds to others and will motivate future work on these states.

The thrust of the reviewer’s comment is curiosity regarding the potential of future positron measurements to elucidate the nature of surface defects in these materials. It would indeed be possible to make a deliberate study of positron annihilation from the surface region of MAPI. Measurements could be performed using implantation energies from 0.25 keV to 2 keV. However, as the reviewer suspects it can be anticipated that these results would be rather complex and challenging to analyse in terms of unambiguous defect assignments. Nevertheless, such studies could be undertaken and a clear picture of the situation would only then emerge. It is highly probable that a fraction of positrons will

re-emit from the surface, form positronium and annihilate, however, it is also possible that positron annihilation at specific vacancy-related defects will be detected and could enable identification. There also exist complementary surface positron methods (positron Auger spectroscopy, etc) and a detailed multi-technique study could be instigated.

2) The referee asks about supercells in the calculations. We have changed the text in the Methods section to make it clear that the 1152 atom calculations were performed only with the computationally efficient but simple atomic superposition method. The full self-consistent first-principles DFT calculations are far more computationally demanding but more accurately describe the physics.

3) The reviewer asks about the inclusion of SOC and the use of hybrid functionals in the calculations. In the revised manuscript we detail test calculations performed including SOC and using hybrid functional for the particular case of the dominant V_{Pb}^{2-} defect. It is found that including SOC reduces the calculated lifetime, but that using hybrid functionals result in an increase in the lifetime and that the contributions are approximately equal leading to cancellation. These results are detailed and discussed at the end of the Methods section (page 18) and in a new section in the Supplementary Information (Theory calculations).

4) “The authors state that a minimum defect density of $\sim 3 \times 10^{15} \text{ cm}^{-3}$ was determined but I would find nice for consistency to have the data for each sample listed as done for other quantities. Could the authors provide all the data in a Table for the different systems in SI?”

We have changed the manuscript as requested and have now included this data in Table 2 and in Tables 1 to 5 in the Supplementary Information. (We used SI units as requested, but for the vacancy concentrations we have used cm^{-3} as is conventional in semiconductor physics – we have given ppm in the Supplementary Information text for comparison.) We also took this opportunity to address uncertainties in the defect concentrations by explaining the details of the calculations. (This also addresses the request from Referee #1, minor comment 2, to include the values used for trapping rates, κ_D). The new text may be found in the SI section ‘Standard Trapping Model defect concentration calculations’ and also at the bottom of page 15.

REVIEWERS' COMMENTS

Reviewer #1 (Remarks to the Author):

I thank the authors for their detailed reply and the revisions made to the manuscript. They clarified in full details and convincingly all points mentioned in my comments. The revised manuscript is in an excellent shape. The important findings and novel approach to experimental identification of point defects in hybrid organic-inorganic perovskites reported in the manuscript form an important milestone both from fundamental and applied perspectives. In short, I recommend publication of this manuscript in its current shape in Nature Communications.

Reviewer #2 (Remarks to the Author):

My comments have been addressed.
Some sentences in the revised manuscript is still confusing,
for example,

For example, the lead vacancy was predicted to that a single shallow acceptor ($-/2-$) level in the gap shallow acceptor as opposed a deep ($0/2-$) acceptor level.

The authors should check the manuscript and SI carefully.

Reviewer #3 (Remarks to the Author):

The authors have satisfactorily clarified the issues raised by the reviewers. Therefore, I suggest the publication of the manuscript in the current form.